

# Effects of plastic ingestion on blood chemistry, gene expression and body condition in wedge-tailed shearwaters (*Ardenna pacifica*)

Nicole Mejia[1,2], Flavia Termignoni-Garcia[1,2], Jennifer Learned[3], Jay Penniman[3] and Scott V. Edwards[1,2]

[1] Department of Organismic and Evolutionary Biology, Harvard University, Cambridge, MA, United States of America
[2] Museum of Comparative Zoology, Harvard University, Cambridge, MA, United States of America
[3] Maui Nui Seabird Recovery Project, Makawao, HI, United States of America

## ABSTRACT

Plastic pollution is a global threat and occurs in almost every marine ecosystem. The amount of plastic in the ocean has increased substantially over the past decade, posing a mounting threat to biodiversity. Seabirds, typically top predators in marine food chains, have been negatively affected by plastic pollution. Here we explored the sublethal effects of plastic ingested by wedge-tailed shearwaters (*Ardenna pacifica*) on the island of Maui, Hawaiʻi. Using analyses of blood chemistry, gene expression, morphometrics and regurgitated stomach contents, we investigated the effects of plastic ingestion on adult wedge-tailed shearwaters from three established colonies. We detected plastic in 12 out of 28 birds; however, we did not find significant relationships between ingested plastic, body condition, gene expression and blood analytes. We found a negative relationship between weight, blood urea nitrogen (BUN), hematocrit and potassium, that could reflect body condition in this population. Genes associated with metabolic, biosynthetic pathways, inflammatory responses, and ribosome function were also upregulated in birds placed in a 'light weight' category. We suggest that upregulated metabolic activity and elevated levels of hematocrit, BUN and potassium in light weight birds might imply dehydration and a response to increased energetic demand from stressors. Repetitive sampling could better inform whether body condition improves throughout the breeding season. We urge researchers to continue using multiple proxies to study effect of plastic ingestion in free-living populations.

# INTRODUCTION

Plastic pollution has been documented across marine ecosystems globally, and an estimated 82-358 trillion pieces of plastic are found afloat in the ocean (*Eriksen et al., 2023*). Worldwide production of plastic has increased nearly 200-fold since the 1950s (*Ritchie, Samborska & Roser, 2023*), and the amount of marine plastic debris has rapidly risen further since 2005 (*Eriksen et al., 2023*). Plastics are composed of durable materials

Corresponding author
Scott V. Edwards,
sedwards@fas.harvard.edu

and estimates of plastic decomposition range up from decades to several hundreds of years (*Barnes et al., 2009*; *Worm et al., 2017*).

The effects of plastic accumulation and persistence in the environment on marine ecosystems are a topic of increasing concern (*Arthur, Baker & Bamford, 2009*; *Hermabessiere et al., 2017*; *Barboza et al., 2020*; *Porcino, Bottari & Mancuso, 2023*). Marine organisms, in particular, are vulnerable to entanglement and ingestion when they come into contact with plastic debris (*Kühn, Bravo Rebolledo & Van Franeker, 2015*; *Ryan, 2018*; *Kühn & Van Franeker, 2020*). A comprehensive review of 747 studies indicated that ingestion of plastics was reported in 701 marine species and entanglement documented in 354 species (*Kühn & Van Franeker, 2020*). In turn, these encounters may present physical impacts to the health of organisms such as blockage and laceration of the digestive tract (*Bjorndal, Bolten & Lagueux, 1994*; *Lazar & Gračan, 2011*; *Charlton-Howard et al., 2023*; *Rivers-Auty et al., 2023*).

Aside from physical consequences, plastic ingestion may lead to physiological impacts by facilitating the transfer of chemicals associated with plastic manufacturing or accumulation of environmental pollutants on their surface (*Chua et al., 2014*; *Rochman et al., 2014*; *Turner et al., 2020*). As plastics degrade, their large surface area-to-volume ratio and hydrophobic surfaces make them effective absorbents for heavy metals and organic chemicals present in the environment (*Verla et al., 2019*). These toxic chemicals, such as polychlorinated biphenyls, bisphenol A, organochlorine pesticides, lead and other heavy metals, have been associated with mutagenic and carcinogenic effects (*Oehlmann et al., 2009*; *Gore et al., 2015*). Plastics, in combination with toxic chemicals, may therefore have detrimental health effects to exposed organisms (*Teuten et al., 2007*; *Pedà et al., 2016*).

Researchers have begun to employ gene expression panels to elucidate the underlying mechanisms driving the observed physiological responses to plastic debris and their toxins (*Rochman, Hentschel & Teh, 2014*; *Granby et al., 2018*; *LeMoine et al., 2018*; *Carrasco-Navarro et al., 2021*; *Patra et al., 2022*). Effects on gene expression include significant expression of liver detoxification enzymes in European seabass (*Dicentrarchus labrax*) that were fed contaminated microplastics (*Granby et al., 2018*) and down-regulation of endocrine associated genes in plastic-exposed Japanese medaka (*Oryzias latipes*, *Rochman, Hentschel & Teh, 2014*). Breeding zebrafish exposed to bisphenol A exhibited disruptions to reproductive processes and changes in expression of DNA methylation enzymes (*Laing et al., 2016*). Another study however, reported no effect of microplastic exposure on zebrafish larva (*Danio rerio*), with most alterations to gene expression disappearing after 14 days (*LeMoine et al., 2018*). It is important to note that the majority of studies investigating physiological effects of environmental pollutants have been conducted in lab-controlled conditions, utilizing fish as model organisms (*Patra et al., 2022*), though gene expression in response to plastic exposure is more difficult to study in wild populations. Further research can help clarify the mechanistic connections between environmentally significant plastic exposure and gene expression in natural populations.

Seabirds are known to ingest marine debris (*Lavers, Bond & Hutton, 2014*; *Provencher et al., 2017*; *Stewart et al., 2020*), and have been used as bioindicators of pollution, including heavy metals and other contaminants (*Vo et al., 2011*; *Espín et al., 2012*; *Lopes et al., 2022*).

Among marine birds, Procellariiforms are at a particular risk to have ingested plastic, often mistaking debris for food (*Sileo et al., 1990*; *Roman et al., 2016*). Ingested debris in marine birds has been linked to mortality (*Roman et al., 2019a*), body mass loss (*Lavers, Bond & Hutton, 2014*) and accumulation of chemical pollutants in tissues (*Tanaka et al., 2020*). However, research on the effects of pollutants from ingested plastic on overall health has produced conflicting results: some studies report impacts to the immune system (*Fernie et al., 2005*; *Costantini et al., 2014*), whereas others detect no changes to the inflammatory response (*Veríssimo et al., 2024*). Yet, in the same study, researchers detected leaching of BDE99, a chemical additive found in plastics, into the brains of yellow-legged/lesser black-backed gulls (*Larus michahellis/Larus fuscus*) that had been fed plastic, as well as reduced activity of enzymes mediating neuromuscular function (*Veríssimo et al., 2024*). Histopathological studies in fledgling seabirds commonly exposed to plastic also present conflicting results; some research describes visible damage to organs with increased plastic load (*Rivers-Auty et al., 2023*), whereas another one did not document chronic damage associated with plastic debris (*Puskic et al., 2024*). One metabolic study reports a link between ingested plastic and effects to growth, calcium, uric acid and cholesterol in flesh-footed shearwaters (*Ardenna carneipes*, *Lavers, Hutton & Bond, 2019*). However, it is unclear whether these patterns are caused primarily by birds experiencing malnutrition rather than as a direct toxicological effect of ingested plastic (*Roman et al., 2021b*). These confounding results highlight the challenges in understanding the severity of effects plastic pollution. Currently, there is a growing consensus that the impacts of ingested plastic vary by species and may depend on specific circumstances (*Bucci, Tulio & Rochman, 2020*). By studying physiological conditions in free-living organisms, we can continue to understand the impacts of plastic debris.

Knowledge gaps and questions remain about the intrinsic aspects of plastic, severity of impact on human health and marine organisms, effective mitigation measures, and biomagnification across the food webs (*Bonanno & Orlando-Bonaca, 2018*; *Galloway et al., 2020*). Previous studies have used a series of techniques to try to understand the effects of plastics on organisms including morphometric studies (*Szabo et al., 2021*), genomic tools (*Laing et al., 2016*; *Granby et al., 2018*; *LeMoine et al., 2018*), or metabolite panels (*Lavers, Hutton & Bond, 2019*). To further our understanding on the effects of plastic pollution, it is important to combine these tools and apply them to free-living populations. Therefore, the aim of this project is to quantify effects of plastic ingestion on physiological function in free-living seabirds known to ingest and otherwise encounter plastic in their natural environment.

In this study, we focused on identifying possible effects of ingestion of plastic debris on gene expression, morphometrics, and blood analytics in three established colonies of wedge-tailed shearwaters (*Ardenna pacifica*, Fig. 1A). Wedge-tailed shearwaters are highly pelagic seabirds that range across the tropical and subtropical areas of the Pacific and Indian Ocean (*Adams, Felis & Czapanskiy, 2020*). They exhibit feeding behaviors, such as contact dipping and surface-seizing (*Adams, Felis & Czapanskiy, 2020*), that increase their susceptibility to plastic ingestion (*Fry, Fefer & Sileo, 1987*; *Roman et al., 2019a*). Both ingestion of plastic (*Fry, Fefer & Sileo, 1987*; *Kain et al., 2016*) and entanglement
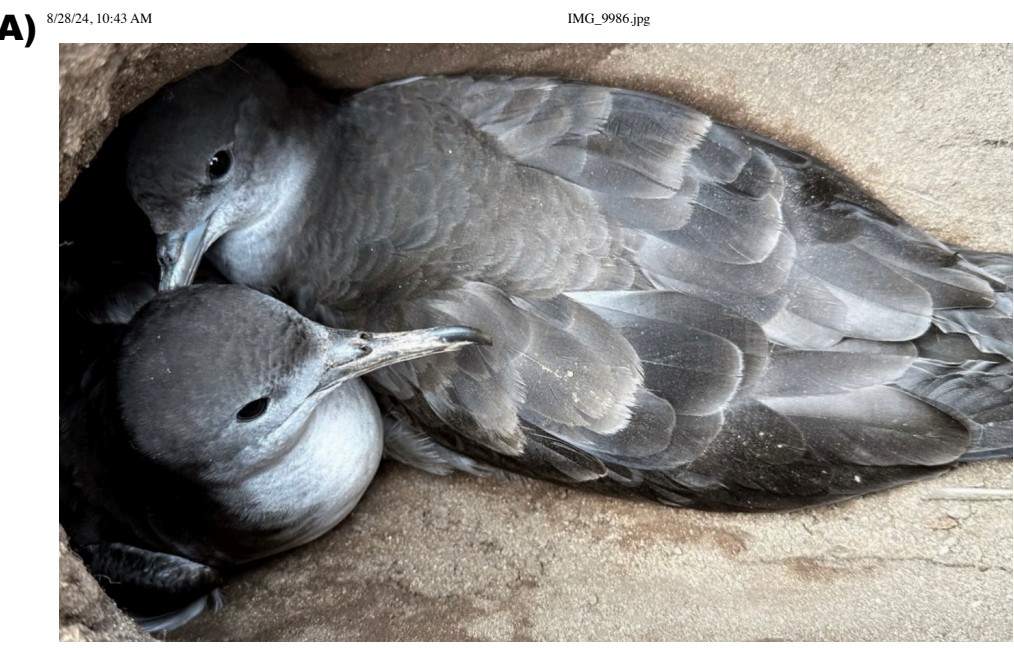

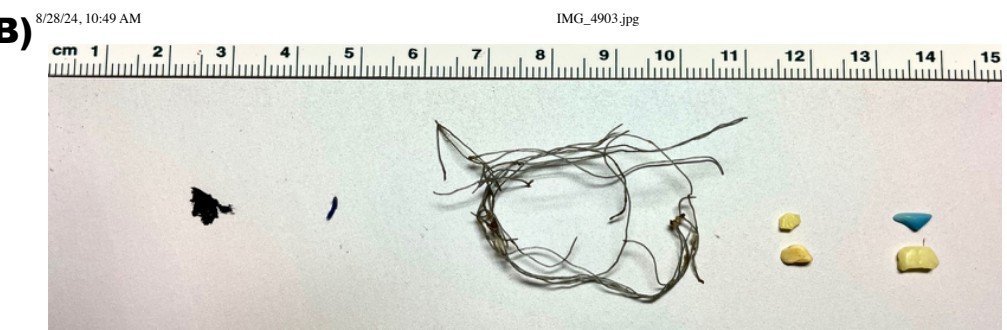

**Figure 1 Sampling information.** (A) Wedge-tailed Shearwater *(Ardenna pacifica)* in Hawaiʻi. Photo credit: Jennifer Learned). (B) Plastic samples. Plastic collected from stomach samples from WTSH.

(*Hyrenbach et al., 2020*) have been previously recorded in wedge-tailed shearwaters on the Hawaiian Islands. Plastic ingestion has also been recorded in wedge-tailed shearwaters chicks and fledglings (*Kain et al., 2016*).

We aimed to investigate wedge-tailed shearwaters on Maui, Hawaiʻi, an island with significant concentration of floating plastic debris in the surrounding waters (*Cózar et al., 2014*). In addition to collecting several morphometric measurements such as weight and blood chemistry, we used transcriptomic data to characterize the activity of genes expressed in whole blood under conditions of plastic ingestion. Given the previous reported effects of

plastic on digestive function in marine organisms (*Bjorndal, Bolten & Lagueux, 1994*; *Pierce et al., 2004*; *Lavers, Bond & Hutton, 2014*; *Ryan, de Bruyn & Bester, 2016*), we anticipated that there would be physiological alterations related to metabolic function in individuals found to have ingested plastic. We also predicted that individuals would differ in whole blood gene expression in ways that might reflect body condition, depending on whether they were found to have ingested plastic.

## MATERIALS AND METHODS

Portions of this text were previously published as part of a preprint (https://www.biorxiv.org/content/10.1101/2022.11.26.517527v1.full.pdf).

### Study site

Sample collection took place in June 2021, before the wedge-tailed shearwaters egg-laying period (mid-June to mid-July), on the island of Maui, Hawai'i. Sampling sites were chosen based on known established colonies of wedge-tailed shearwaters. Although all of the colonies are in protected areas, the three chosen sites were situated on the seashore near frequently visited beaches. Each of the sites was visited over a two-day period within a two-week span in June 2021. Kamaole Park III (20°42′36.72″N, 156°26′43.8″W) was visited on June 9 and 10, Ho'okipa Beach Park (20°56′7.8″N, 156°21′19.8″W) on June 15 and 16, and Hawea Point (21°0′28.0794″N, 156°39′53.9994″W) on June 21 and 22. Sampling was conducted in the evening, starting at around 8 pm HST and ending at around 10 pm HST. In total, we captured and processed 28 birds as they returned from feeding. There were seven birds sampled from Kamaole Park III, ten birds sampled from Ho'okipa Beach Park, and 11 birds sampled from Hawea Point. Field experiments were approved by the Hawaii Division of Forestry and Wildlife (permit number: 08487).

### Blood sampling

Blood chemistry can be used as an indicator of overall health and morphometric data are widely used in ecological studies to estimate body condition (*Harr, 2002*; *Mallory et al., 2010*; *Labocha & Hayes, 2012*). We began processing each bird by first collecting blood samples to mitigate the physiological, chemical and expression effects of handling-induced stress responses. Blood samples were collected from 28 birds for gene expression analysis, and from 25 birds for blood chemistry analysis. Blood samples from four birds were insufficient for blood chemistry analysis and a blood sample from one bird was insufficient for gene expression analysis (summarized in Table 1). Additionally, we collected a blood sample for gene expression from one bird that had independently been transported to the field station for medical attention. Although we analyzed the samples collected from this individual, we lacked information about its colony of origin.

We collected approximately 200 µl of blood using a syringe from the medial metatarsal vein and used styptic powder to stop bleeding when necessary. We added 100 µl of the blood to a vial containing RNAlater buffer for gene expression analysis, and 20 µl of blood in heparin tubes for iStat cartridge analysis. The samples were processed for blood chemistry analysis in the field using the iStat analyzer at the end of each collecting bout. The iStat

**Table 1  Bird ID and data available for each specimen.**

| Bird ID | Sex | Plastic | Description of contents | Blood sample | Location of sampling |
|---------|-----|---------|-------------------------|--------------|----------------------|
| N001 | F | NA | NA | Gene only | Unknown* |
| N002 | F | (+) | 10 squid beaks<br>1 black hard fragment | Both | Kamaole Park III |
| N003 | F | (-) | 1 squid beak and fish | Gene only | Kamaole Park III |
| N004 | NA | (+) | 1 squid beak<br>1 yellow hard fragment<br>1 fiber string | Chem only | Kamaole Park III |
| N005 | M | (+) | 1 blue hard fragment<br>1 brown hard fragment<br>2 pink –orange hard fragment | Both | Kamaole Park III |
| N006 | F | (-) | Squid and 2 squid beaks | Both | Kamaole Park III |
| N007 | F | (+) | 2 black hard fragments | Both | Kamaole Park III |
| N008 | F | (-) | 2 fish spines and squid | Both | Kamaole Park III |
| N009 | F | (-) | No identifiable food items | Both | Hoʻokipa |
| N010 | M | (+) | Squid pieces<br>2 squid beaks<br>Fishing line bundle | Both | Hoʻokipa |
| N011 | M | (+) | 1 black hard fragment<br>4 squid and fish spines | Both | Hoʻokipa |
| N012 | F | (-) | 4 squids | Both | Hoʻokipa |
| N013 | F | (-) | No unidentifiable food items | Both | Hoʻokipa |
| N014 | M | (-) | Fish | Both | Hoʻokipa |
| N015 | F | (+) | Fish and 1 squid beak<br>2 yellow hard fragments | Both | Hoʻokipa |
| N016 | M | (-) | Fish | Both | Hoʻokipa |
| N017 | M | (+) | 1 squid beak<br>1 brown hard fragment | Both | Hoʻokipa |
| N018 | F | (-) | 4 squid pieces | Both | Hoʻokipa |
| N019 | M | (-) | No identifiable food pieces | Both | Hawea |
| N020 | F | (+) | 5 hard fragments<br>40 squid beaks | Both | Hawea |
| N021 | M | (-) | Squid | Both | Hawea |
| N022 | F | (-) | Squid<br>4 squid beaks | Both | Hawea |
| N023 | F | (-) | 1 squid beak | Both | Hawea |
| N024 | F | (+) | 4 squid beaks<br>1 green hard fragment | Gene only | Hawea |
| N025 | F | (+) | Fish<br>1 yellow hard fragment<br>1 blue hard fragment | Both | Hawea |
| N026 | M | (-) | No identifiable food parts | Both | Hawea |
| N027 | M | (-) | 1 squid beak<br>Rock | Both | Hawea |

Table 1 (*continued*)

| Bird ID | Sex | Plastic | Description of contents | Blood sample | Location of sampling |
|---------|-----|---------|------------------------|--------------|----------------------|
| N028 | F | (+) | 1 black hard fragment | Both | Hawea |
| N029 | F | (−) | No identifiable food parts | Gene only | Hawea |

**Notes.**

An ID was appointed for each specimen. The sex is indicated for each specimen. Plastic refers to if that particular specimen had ingested plastic; (+) indicates presence of plastic and (-) indicated that that specimen did not have ingested plastic. Blood sample refers to what information was obtained from the collected blood. Description of the stomach contents is also included. Specimens where only genetic information is available is denoted by "Gene only". Specimens where only a blood chemistry panel is available is denoted by "Chem only". Specimens where both genetic information and blood chemistry panels are available is denoted by "both". Location of sampling indicates the location of the beach where sampling took place. Sampling was conducted at three locations of known established wedge-tailed shearwaters colonies: Kamaole Park III, Hoʻokipa Beach Park and Hawea Point. N001 bird was brought to the station for medical attention. A blood sample was collected for this bird, but no other measurements were taken and therefore not included in analyses.

Chem8+ cartridge provided us with the following blood analytics; sodium (Na mmol/L), potassium (K mmol/L), chloride (Cl mmol/L), ionized calcium (iCa mmol/L), total carbon dioxide (TCO2), glucose (Glu mg/dL), urea nitrogen/urea (BUN mg/dL), creatinine (Crea mg/dL), hematocrit (Hct %PCU), hemoglobin (Hb g/dL), anion gap (AnGap mmol/L). Blood samples in RNAlater buffer were stored at −20 °C for two months before being express shipped on dry ice to Cambridge, MA for processing.

## Morphometric data

We measured tarsus length, bill length, nares depth and width using calipers, wing chord length using a ruler, and weight using a Pesola scale ($n = 28$). Each bird was banded, and the band number was recorded if the bird was a recapture.

## Gastric lavage

We followed the procedure described by *Duffy & Jackson (1986)* for gastric lavage to collect potentially ingested plastic. The method outlined by *Duffy & Jackson (1986)*, involves a stomach pump system in which the seabird is filled with ambient-temperature seawater through a lavage tube and then inverted over a bowl to promote and collect regurgitation. This procedure was carried out twice for each bird ($n = 28$). Before being released back into the colony, birds were inspected for bleeding or other signs of distress. Stomach contents were strained through a one mm sieve and contents were examined for presence of plastic. Each bird was assigned presence or absence of plastic accordingly (Table 1). The Standing Committee on the Use of Animals and Teaching at Harvard University provided full approval for this research (project number 24-06).

## DNA purification and polymerase chain reaction

We used the Qiagen QiAMP DNA Blood kit for DNA purification preceding amplification *via* to the polymerase chain reaction (PCR).

We used the universal method described by *Fridolfsson & Ellegren (1999)* for sexing in birds with PCR reaction. The two-primer system is as follows:

2550F: 5′-GTTACTGATTCGTCTACGAGA-3′

2718R: 5′-ATTGAAATGATCCAGTGCTTG-3′

Using this primer system, we employed standard PCR on the templates of DNA extracted from unknown-sex *A. pacifica* species. The PCR mixture (15 µl) contained 1.5 µl of 10X

buffet, 0.5 μl of dNTP (10 pmol), 0.5 μl of forward primer (10 pmol), 0.5 μl of reverse primer (10 pmol), 0.1 μl of NEB Ta1 (5U/uL), and 9.4 μl of H2O. 2.5 μl of the DNA extraction was used. The PCR program was as follows: 94 °C for 5 min, 94 °C for 30 s, 60 °C for 30 s *touchdown, −1.0 °C/cycle × 10 cycles, 72 °C for 30 s, 94 °C for 30 s, 50 °C for 30 s × 30 cycles, 72 °C for 30 s, 72 °C for 5 min, 4 °C hold. We used molecular grade $H_2O$ as a negative control. A negative control was essential for possible misinterpretation due to contamination or other factors. We separated the PCR product through electrophoresis on a 2% agarose gel at 90 V for about 1 h and stained the gel with *GelRed*™.

## RNA isolation and sequencing

Before initiating RNA isolation, RNAlater was removed from the mixture of tissue and buffer by centrifuging aliquots at 20,800 × g (RCF). We removed supernatants from the pellets immediately incorporated Qiazol and zirconia/silica 1 mm beads (BioSpec Products) for homogenization on the TissueLyser LT (Qiagen, Hilden, Germany), followed by the RNeasy Plus Universal mini kit protocol (Qiagen, Hilden, Germany). A KAPA mRNA Hyperprep kit and a NOVASeq SP platform was used at the Harvard Bauer Sequencing Core Facility to sequence paired end reads of 150 bp length, yielding between 20 and 30 million reads per sample.

## Data analysis

Using R v. 3.5.1 (*R Core Team, 2018*), we explored relationships between blood analytes, presence of plastic and sex using t-tests, principal component analysis (PCA), logistic linear models (GLM) and conducted a Shapiro–Wilk test of normality. *T*-tests were used to compare the mean differences in blood analytes between birds that had ingested plastic and those that did not. Each of the blood analytes was analyzed independently. Differences were considered statistically significant when $p < 0.05$. We conducted PCA analysis to assess whether the categories of plastic ingestion and sex cluster based on morphometric, blood analyte, and genetic data. We initially conducted a PCA analysis to account for potential sex-related differences. Additional PCA analysis tested whether individuals clustered together based on physiological differences attributed to plastic ingestion. Finally, we used a logistic linear regression to study the association between the variables. Differences were considered statistically significant when $p < 0.05$.

We assessed RNA quality using RNA integrity number (RIN) values and estimated sequencing quality using Phred Scores (*Ewing & Green, 1998a*; *Ewing et al., 1998b*). RIN values assign a numerical value to the quality of the RNA that we analyzed by evaluating the integrity of 18S and 28S rRNSs (*Puchta, Boczkowska & Groszyk, 2020*). A RIN value of 8 or higher indicates higher RNA quality and integrity, whereas values below 5 indicate varying degrees of RNA degradation. Phred scores evaluate the quality of sequences; higher values (90% and above) denote better quality sequences.

We aligned Illumina sequence reads to the publicly available reference genome of Cory's Shearwater (*Calonectris borealis,* NCBI accession number PRJNA545868, *Feng et al., 2020*) with the RNA sequence mapper STAR (Spliced Transcripts Alignment to a Reference) (*Dobin et al., 2013*), followed by transcript quantification in R with RSEM (*Li & Dewey,*

*2011*) and differential gene expression analysis with DESeq2 (*Love, Huber & Anders, 2014*). We created heatmaps to visualize patterns of differential gene expression between birds of differing plastic status. We conducted analyses with weight as the independent variable. We divided weight into three factors—lower third, medium, and upper third—by dividing the range of weight into intervals according to its observed distribution. Weights were categorized into the following ranges: 296–361 g for low weight birds ($n = 6$); 366–421 g for medium weight birds ($n = 16$); and 446–496 grams for heavier birds ($n = 6$). The three wedge-tailed shearwaters colonies were treated as one population in the analyses of gene expression, because we do not expect there to be significant genetic divergence between these three geographically close colonies (*Herman et al., 2022*). In all of the tests, we considered a *p*-value of <0.05 as statistically significant and also reported relationships with *p*-values <0.1, given the relatively small sample sizes examined. A summary of the statistical tests conducted can be found in Tables S1, and S2.

We conducted gene ontology (GO) analysis in R using the package gprofiler2 with *Gallus gallus*, *Taeniopygia guttata* and *Mus musculus* as the model systems in the search database (*Kolberg et al., 2020*). We used ggplot2 and plotly for plotting as outlined in *Kolberg et al. (2020)*. We separated terms into Gene Ontology, KEGG pathways and Reactome databases (*Kanehisa & Goto, 2000*; *Kanehisa, 2019*; *Kanehisa et al., 2023*; *Milacic et al., 2024*).

# RESULTS

## Sex determination through PCR

PCR amplification of the sex of each bird showed that our sample consisted of 10 males and 18 females. Bird N001 was a female, but neither stomach samples nor blood analytes were not collected for this bird (Table 1). The sex of one bird, N004, remained unknown due to insufficient amount of blood to carry out analyses.

## Stomach contents from gastric lavage

Plastic was found in 12 of the 28 birds sampled for plastic (see Fig. 1B, Table 1 and Fig. S1). These included fishing line ($n = one\ bundle$), fiber ($n = one\ fiber$) and hard fragments of plastics ($n = 21$ pieces). Hard fragments of plastic ranged in color, from orange-pink to yellow to black. The majority of birds found with ingested plastic were found to have one to two plastic pieces, with bird N005 having the most (five hard fragments, Table 1). Bird N010 was found to have one large bundle of fishing line (see Fig. 1B and Table 1). Blood sample type, sex, presence and description of plastic for each bird sampled can be found in Table 1. Four out of the 10 identified males contained plastic. Seven out of 17 identified females contained plastic. Bird N004 of unknown sex also contained plastic (Table 1).

## Blood analytics, morphometric and stomach contents

Although there were differences between the averages of the blood analytes of birds with plastic and birds without ingested plastic, the differences were not significant ($p > 0.05$, see Fig. S2, Tables S2 and S3). However, birds that had ingested plastic weighed more (mean = 407.13 g) than birds that had not ingested plastic (mean = 377.08 g). Previous studies report an average body mass of 375 g in male shearwaters and 381 g in female

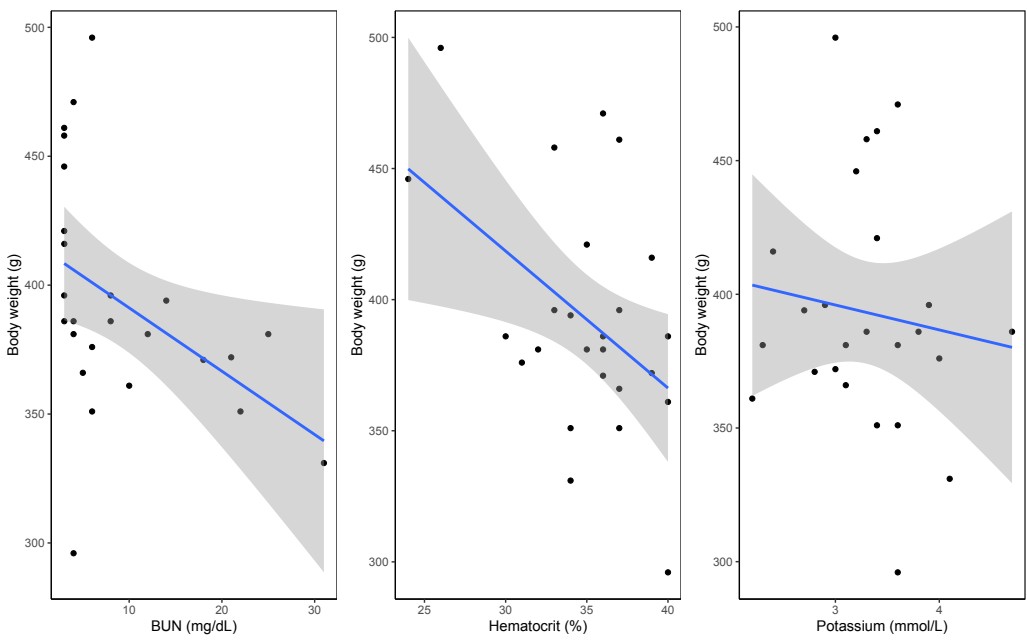

**Figure 2** **Significant results from general linear model of blood analytes with weight as the variable.** (A) Hematocrit as percentage with weight as variable. (B) Urea nitrogen/urea with weight as the variable. (C) Potassium with weight as the variable.

wedge-tailed shearwaters (*Totterman, 2015*). Birds that had ingested plastic also had lower levels across all of the blood analyte panels ($n = 10$) except chloride levels, but these did not show a significant effect ($p > 0.05$).

### Relationships between blood analytics, morphometric and stomach contents

A Shapiro–Wilk test of normality indicated that body mass in relation to plastic exposure was non-parametric. Linear models indicated a significant negative relationship between blood urea nitrogen (BUN) and body mass (estimate $= -0.01$, $p = 0.001$). Similarly, hematocrit (estimate $-0.0175$, $p = 0.001$) and potassium (estimate $= -0.04$, $p = 0.04$) showed significant effects, with higher levels associated with lower body mass (see Fig. 2, Table 2 and Table S2). Near-significant negative effects were observed between TCO2 levels and body mass (estimate $= 0.0178$, $p = 0.06$) and between body mass and presence of plastic (estimate $= -0.02$, $p = 0.08$, see Fig. S3). Residual deviance (30.22) suggests a good fit of the model to the data. No other significant effects were observed between the presence of plastic, blood chemistry panels and morphometric measurements including relationship between body mass and plastic presence ($p < 0.05$). A summary of the model outputs, sample number, variables and tests can be found in Table 2 and Table S2.

Our PCA results did not reveal distinct patterns or clustering of birds based on blood chemistry or morphometric measurements when analyzed by sex or the presence of plastic (Figs. S4 and S5). Specifically, we did not observe clear separation by sex in the

**Table 2  Summary of tests and outcomes.**

| Dependent variable | Independent variable | Test | Sample size | p-value |
|---|---|---|---|---|
| Sodium (mmol) | Presence of plastic | T-test | Plastic $n = 11$<br>No Plastic $n = 14$ | 0.64 |
| Potassium (mmol) | Presence of plastic | T-test | Plastic $n = 11$<br>No Plastic $n = 14$ | 0.87 |
| | Body weight | General linear model | Plastic $n = 11$<br>No Plastic $n = 14$ | 0.04* |
| Chloride (mmol) | Presence of plastic | T-test | Plastic $n = 11$<br>No Plastic $n = 14$ | 0.09 |
| Ionized Calcium (mmol) | Presence of plastic | T-test | Plastic $n = 11$<br>No Plastic $n = 14$ | 0.65 |
| Total Carbon Dioxide (mmol) | Presence of plastic | T-test | Plastic $n = 11$<br>No Plastic $n = 14$ | 0.06 |
| | | General linear model | Plastic $n = 11$<br>No Plastic $n = 14$ | 0.06 |
| Glucose (mg/dL) | Presence of plastic | T-test | Plastic $n = 11$<br>No Plastic $n = 14$ | 0.65 |
| BUN (mg/dL)<br>*Blood urea nitrogen* | Presence of plastic | T-test | Plastic $n = 11$<br>No Plastic $n = 14$ | 0.66 |
| | Body weight | General linear model | Plastic $n = 11$<br>No Plastic $n = 14$ | $1.81*10^{-3}$ |
| Creatine (mg/dL) | Presence of plastic | T-test | Plastic $n = 11$<br>No Plastic $n = 14$ | 0.34 |
| Hematocrit (%) | Presence of plastic | T-test | Plastic $n = 11$<br>No Plastic $n = 14$ | 0.62 |
| | Body weight | General linear model | Plastic $n = 11$<br>No Plastic $n = 14$ | $1.27*10^{-3}$ |
| Hemoglobin (g/dL) | Presence of plastic | T-test | Plastic $n = 11$<br>No Plastic $n = 14$ | 0.34 |
| Anion Gap (mmol) | Presence of plastic | T-test | Plastic $n = 11$<br>No Plastic $n = 14$ | 0.34 |
| Weight (g) | Presence of plastic | T-test | Plastic $n = 11$<br>No Plastic $n = 14$ | 0.08 |
| | Sex | | Females = 18<br>Males = 10 | 0.44 |
| | | General linear model | Plastic $n = 11$<br>No Plastic $n = 14$ | 0.09 |
| | | Shapiro–Wilk Test | Plastic $n = 11$<br>No Plastic $n = 14$ | $W = 0.94$<br>p-value = 0.13 |
| Morphometric measurements | Presence of plastic | Principal Component Analysis | Plastic $n = 12$<br>No plastic $n = 16$ | - |
| | Sex | | Female $n = 17$<br>Male $n = 10$<br>*1 Unknown not included | |

**Table 2** (*continued*)

| Dependent variable | Independent variable | Test | Sample size | *p*-value |
|---|---|---|---|---|
| Blood Chemistry | Presence of plastic | Principal Component Analysis | Plastic $n = 11$ <br> No Plastic $n = 14$ | – |
| | Sex | | Female = 14 <br> Male = 10 <br> *1 Unknown not included | |
| Gene expression | Presence of plastic | Differential Gene Expression | Plastic $n = 11$ <br> No plastic $n = 16$ <br> *2 Unknown | – |
| | Sex | | Female $n = 18$ <br> Male $n = 10$ <br> Female plastic = 7 <br> Female no plastic = 10 <br> Male plastic = 4 <br> Male no plastic = 6 | – |

**Notes.**
The dependent variable column lists all of the variables tested in relation to the independent variable listed in the column to the right. The test(s) conducted on those relationships and the outcome of that test is listed next to the variables tested. The sample size for each of the tested groups is also listed. A more detailed model output is detailed in Table S2. Significant values are marked with an asterisk (*).

morphometric data, despite evidence of sexual dimorphism in wedge-tailed shearwaters (*Totterman, 2015*).

## RNA-seq analyses

The RIN value used to assess RNA integrity of each was approximately 8.4–6 for most samples (Table S4). Sample N005 had a lower RIN value of 4.8. Phred scores were used to assess sequencing quality (Fig. S6). On average, all samples reached phred scores of 30 or greater, indicating good quality for downstream analyses.

Figure S7 shows the number of reads per sample, which ranged from 30 million to 80 million. The proportion of reads mapping to multiple locations in the reference genome ranged from 1.3% to 2.7% per sample (Fig. S8). In contrast, uniquely mapped reads accounted for more than 40% of the total, and some samples achieved higher mapping rates of 60% (Fig. S9). The high proportion of uniquely mapped reads indicated efficient mapping to a closely related reference genome for our species of interest. However, samples with less than 60% unique mapping may be affected by RNA degradation from blood samples (*Dobin & Gingeras, 2015*). Recovering more than 60% of the reads might be challenging because we did not use a reference genome from the same species for the transcriptome alignment.

Fourteen genes differentiated males with ($n = 4$) and without plastic ($n = 6$), (Fig. S10A). Four genes differentiated females with ($n = 7$) and without plastic ($n = 10$) (Fig. S10B). When conducting a separate analysis, using sex as the main variable, eleven genes differentiated males and females with plastic ($n = 11$, Fig. S11A). Forty-three genes differentiated males and females without plastic ($n = 16$, Fig. S11B).

Heavier birds exhibited a downregulation in the expression of 18 significantly differentiated genes compared to birds in the other two categories (Fig. 3A, log2FC = $-0.59$). Lighter birds showed an upregulation of the expression of the same genes. The top two genes responsible for body mass differentiation, Ankrd11_1 and Hsph 1 (Fig. 3C), were

significantly upregulated in heavier birds ($p < 0.05$, log2FC = 0.74). Genes upregulated in heavier birds were associated with trimethylation and cell cycle function (Fig. S12B). The top twelve genes upregulated in lighter birds (Fig. 3A) were associated with several metabolic and biosynthetic processes, and ribosome function (see Fig. 3B and Fig. S12B).

## DISCUSSION

In this study we investigated the possible effects of ingestion of plastic debris in wedge-tailed shearwaters on Maui, Hawaiʻi by analyzing morphometric measurements, stomach contents and blood samples. We detected ingested plastic in 12 out of 28 sampled birds. However, we did not find statistically significant relationships between presence of ingested plastic and either body condition or gene expression panels. We found statistically significant relationships between body conditions and some blood analytes, including a negative relationship with hematocrit, BUN, and potassium. Analyses of gene expression also suggested differences in expression levels of various genes between lighter and heavier birds. Although these findings do not suggest an effect of plastic ingestion on body condition ($p > 0.05$), they may inform us on the overall body condition of this seabird population.

Differences in gene expression among categories of body mass were attributed to two key patterns (i) upregulation of biosynthetic and metabolic pathways in lighter birds and (ii) downregulation of biosynthetic pathways in heavier birds. Analysis of differentially expressed (DE) genes revealed an upregulation of genes involved in biosynthetic processes in lighter birds, which are essential for accumulation of body mass. Additionally, birds in the low-weight category exhibited upregulation of genes involved in metabolic processes of organonitrogen compounds (compounds comprised of nitrogen atoms). Our linear model revealed a relationship between lighter birds and higher blood urea nitrogen (BUN) levels. BUN levels have been used as an indicator of dehydration in birds; elevated levels can imply dehydration (*Harris, 2009*). Higher levels of urea and uric acid have also been associated with removal of protein stores from muscle (*Alonso-Alvarez et al., 2002*; *Ferrer et al., 2013*) and have been reported in a breeding population of brown skuas (*Stercorarius antarcticus*) experiencing low breeding success (*Graña Grilli, Pari & Ibañez, 2018*). Other studies monitoring body condition throughout the breeding season have noted that changes in body mass and blood metabolite may be sex-specific; for example, body condition can improve late in the breeding season (*Graña Grilli, Pari & Ibañez, 2018*; *Fitzgerald, Lynch & Jessopp, 2022*; *Lerma et al., 2022*). Recapturing individual wedge-tailed shearwaters later in the breeding season would provide more insight into whether higher BUN levels indicate poor body condition or sex-specific variation in nutritional status.

Potassium and hematocrit can also serve as indicators of body condition. Whereas research on potassium levels in plasma has mostly focused on poultry, it has nonetheless emphasized the key role of potassium in maintaining an important role in energy metabolism, muscle function and hydration (*Leach et al., 1959*; *Oliveira et al., 2005*; *Araujo et al., 2022*). Tests for potassium can therefore likely be used to diagnose dehydration and kidney related problems; here higher levels of potassium might indicate possible
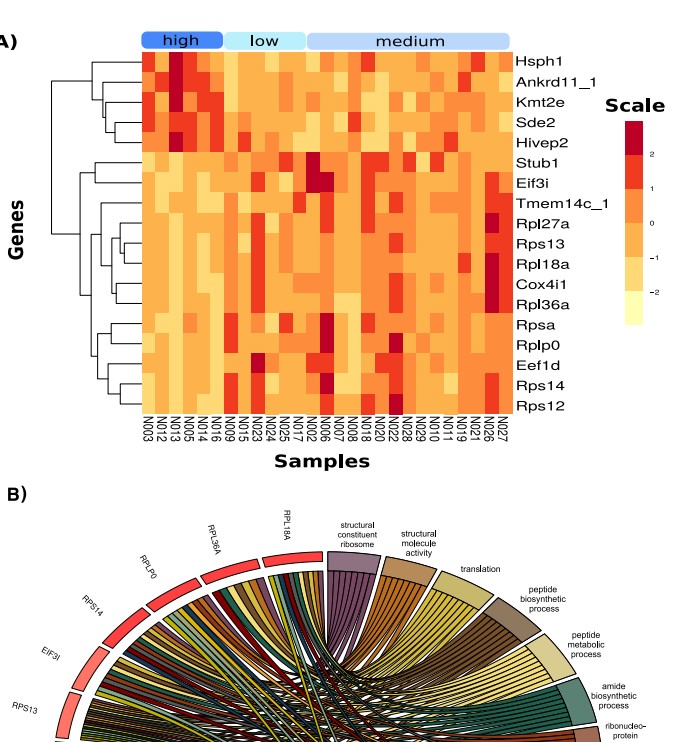

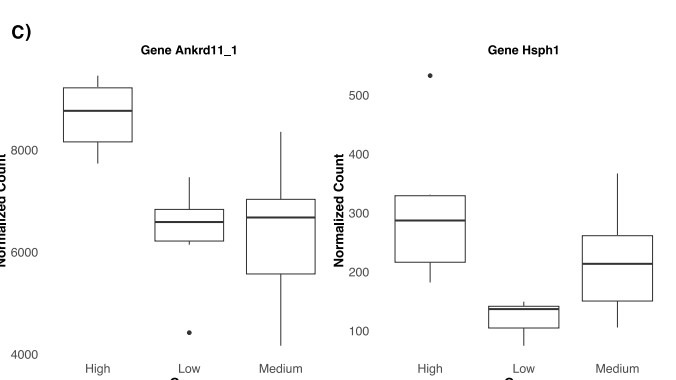

**Figure 3  Differential Expression of genes with three categories of weight; low, medium and high.** (A) Heatmap showing the 18 significantly DE genes. (B) Chord diagram of gene ontology terms. This plot visualizes the relationship between key differentially expressed genes in low and higher weight birds and their associated Gene ontology (GO) terms. A significant portion of the genes are linked to ribosomal biogenesis and protein synthesis. (C) Boxplots of normalized count in the two top genes showing differences in average counts between the three weight categories.

dehydration (*Simon, Hashmi & Farrell, 2024*). Similarly, increased levels of hematocrit have also been associated with dehydration or tied to metabolic demands (*Hammond et al., 2000*; *Fair, Whitaker & Pearson, 2007*). However, the relationship between hematocrit and individual fitness is not necessarily linear. For example, although increased hematocrit can increase delivery of oxygen, it has also been associated with more viscous blood which can in turn decrease delivery of oxygen to tissues (*Birchard, 1997*; *Schuler et al., 2010*; *Williams, 2012*). A study of grasshopper sparrows (*Ammodramus savannarum*) reported a decrease in body fat and an increase in hematocrit after birds faced severe weather, suggesting that increased hematocrit was a physiological response to meet metabolic demand (*Freeman et al., 2023*). Our finding of upregulation of genes involved in metabolism in lighter wedge-tailed shearwaters would support this suggestion. Other studies have linked high levels of hematocrit and energy demands during reproduction (*Lownie et al., 2022*) and migration (*Krause et al., 2016*). Given that we conducted sampling at the beginning of the breeding season, we cannot rule out the possibility that increased levels of hematocrit are due to stressors encountered by birds during migration or help meet energy demands at the start of the breeding season. Nonetheless, elevated levels of BUN, potassium, hematocrit, and metabolic activity in lighter birds suggest that some individuals in this population of wedge-tailed shearwaters may be experiencing signs of dehydration. However, levels of hematocrit have been reported to decrease during parental care in several vertebrates (*Williams et al., 2004*; *Fair, Whitaker & Pearson, 2007*; *Hanson & Cooke, 2009*). Therefore, repeated sampling may inform whether the body conditions observed at the start of the season are due to the energy demands of migration and onset of reproduction, or if there are other factors affecting this population.

Ours was an exploratory study to test how to use multiple tools to assess effects on body condition, and it has several limitations. We acknowledge that because we were testing numerous variables, there is the possibility of detecting false positives, because one in every 20 parameters tested has a chance of achieving a *p*-value of significance (*Smith & Ebrahim, 2002*; *Selvin & Stuart, 1966*). Another significant challenge in our study was the possibility of false negative results. The flushing technique for gastric lavage is meant to empty out the proventriculus but may not necessarily empty out the contents of the gizzard in Procellariids (*Duffy & Jackson, 1986*). The stomach in Procellariiformes can be divided into two sections: the proventriculus and the gizzard. When plastic is first ingested, it passes through the large thin-walled proventriculus before it travels down to the gizzard where larger pieces remain until they are grinded down. A lack of plastic content in the proventriculus could mean that it was emptied out faster than from the gizzard (*Nania & Shugart, 2021*). This creates the possibility that, when we sampled for plastic ingestion, plastic had passed through the proventriculus and we did not fully capture the plastic load in every sampled bird. Additionally, the time elapsed between exposure to plastics and blood sampling may dampen the observed effects of plastic on levels of gene transcription (*LeMoine et al., 2018*; *Zhao et al., 2021*). Because we do not know the time or length of exposure to plastic ingestion, we cannot be certain that we are detecting the full extent of plastic exposure and its effect on gene expression. Other methods of sampling for plastics,

such as necropsies, might be more effective at obtaining a more representative plastic load but do not always allow for repeated sampling (*Provencher et al., 2019*).

Our ability to detect a statistically significant relationship may be due to our small sample size of analyzed birds ($n = 28$). For example, other studies on wedge-tailed shearwaters found that male wedge-tailed shearwaters are slightly larger than females across wing, tarsus, bill depth, length, and width measurements (*Totterman, 2015*), a pattern we were not able to detect in our sample. Our small sample could also be the reason that we detected a negative, but not significant relationship between plastic presence and body mass and TCO2 ($p < 0.1$). *Provencher, Bond & Mallory (2015)* suggest 100 individuals per site per year to determine proper sample of ingested plastic in Procellariiformes; we did not reach this sample size. However, analyzing transcriptomes or other assays for each of over 100 birds could also be prohibitively expensive. Finally, we focused on the effect of the presence or absence of plastic, rather than plastic load. It might be important to consider both the load and presence/absence in future analyses of gene expression, as is suggested in other literature on plastic pollution (*Provencher et al., 2019*). Furthermore, the plastic loads we detected (see photo, Fig. 1C and Table 1) may not be large enough to elicit a strong physiological response.

The relationship between body weight and plastic in birds has been inconsistent over the years (*Lavers, Hutton & Bond, 2021*). Much research has detected no relationship between ingested plastic and body weight (*Sievert & Sileo, 1993*; *Cousin et al., 2015*); negative relationships (*Spear, Ainley & Ribic, 1995*; *Lavers, Bond & Hutton, 2014*); and positive relationships (*Puskic et al., 2024*). Feeding experiments to chicks (*Gallus gallus*, *Ryan, 1988*); and Japanese quail chicks (*Coturnix japonica*) fed plastic (*Roman et al., 2019b*) has impacted growth, but the effects disappeared once the Japanese quail chicks reached adulthood. A negative relationship between body mass and plastic presence might be more complex to decipher in this population, because researchers have noted that if an individual is already experiencing poor body condition, they may experience reduced prey discrimination (*Roman et al., 2021a*). Therefore, it is difficult to conclude if seabirds that had ingested plastic were already experiencing poor body condition. Additionally, ornithologists debate which morphometrics provide the best estimates of body condition, and some literature recommends gathering multiple proxies to fully understand body condition in seabirds (*Mallory et al., 2010*; *Labocha & Hayes, 2012*). A major challenge is the absence of control groups in free-living populations, which along with a larger sample size could help disentangle effects from plastic from those of other stressors. Despite the inherent limitations of studying free-living populations, the ambiguous findings here highlight the importance of continuing to research effects of plastic ingestion on free-living populations using multiple proxies. Additionally, our study is, to our knowledge the only study documenting patterns of gene expression in the context of plastic ingestion in natural populations of birds of any kind.

## CONCLUSION

We did not detect a significant relationship between the presence of plastic and body condition as measured by blood chemistry in the wedge-tailed shearwaters population

on Maui, Hawaii. There was a negative relationship between presence of plastic, body mass and TCO2, but it was not statistically significant. We detected significant negative relationships between body mass and BUN, potassium and hematocrit all of which may point to signs of dehydration in some individuals in this population. Genes involved in metabolic pathways were also upregulated in lighter birds. We suggest that higher metabolic activity, elevated BUN and hematocrit levels could also indicate utilization of body stores to sustain energetic demand from stressors. Repeating sampling later in the breeding season and larger sample sizes would help clarify whether body condition at the time of sampling is a result of energy demands from migration, onset of reproduction or other factors. Despite the limitations of this study, we encourage researchers to continue using multiple proxies to fully understand body condition in free-living populations. To our knowledge, this is the first study that combines analyses of gene expression, blood analyte panels and morphometric measurements to assess the effect of plastic ingestion in a free-living population of seabirds. Incorporating larger sample numbers, and types and loads of plastic into analyses of gene expression, as well as repeat and non-destructive sampling might provide a more comprehensive account not only of how plastic ingestion and other stressors affect marine life.

## ACKNOWLEDGEMENTS

We are grateful to Kallalei Ryden, Cecelia Frisinger and the staff at Maui Nui Seabird Recovery Project for their assistance in collecting samples. We would like to thank Dr. Paul McCurdy and Jeremiah Trimble for logistical support; also, to Dr. Andrés Cózar for granting access to data on plastic concentrations around the island of Maui. Thank you to Lauren Roman, two anonymous reviewers and Heidi Aumun for their insightful comments and suggestions on the manuscript.

### Funding

This work was supported by the Harvard College Office of Undergraduate Research and Fellowships and the Harvard University Museum of Comparative Zoology. The Wetmore Colles Fund of the Museum of Comparative Zoology at Harvard funded the open access charges, article processing fees and the graphical abstract. The funders had no role in study design, data collection and analysis, decision to publish, or preparation of the manuscript.

### Grant Disclosures

The following grant information was disclosed by the authors:
Harvard College Office of Undergraduate Research.
Fellowships and Harvard University Museum of Comparative Zoology.
Museum of Comparative Zoology at Harvard.

### Competing Interests

Scott Edwards is an Academic Editor for PeerJ. The authors declare that they have no other competing interests.

## Author Contributions

- Nicole Mejia conceived and designed the experiments, performed the experiments, analyzed the data, prepared figures and/or tables, authored or reviewed drafts of the article, and approved the final draft.
- Flavia Termignoni-Garcia performed the experiments, analyzed the data, prepared figures and/or tables, authored or reviewed drafts of the article, and approved the final draft.
- Jennifer Learned conceived and designed the experiments, performed the experiments, prepared figures and/or tables, authored or reviewed drafts of the article, and approved the final draft.
- Jay Penniman performed the experiments, authored or reviewed drafts of the article, and approved the final draft.
- Scott V. Edwards conceived and designed the experiments, prepared figures and/or tables, authored or reviewed drafts of the article, and approved the final draft.

## Animal Ethics

The following information was supplied relating to ethical approvals (i.e., approving body and any reference numbers):

The Harvard University Faculty of Arts and Science Standing Committee on the Use of Animals and Teaching provided full approval for this research (project number 24-06).

## Field Study Permissions

The following information was supplied relating to field study approvals (i.e., approving body and any reference numbers):

Field experiments were approved by the Hawaii Division of Forestry and Wildlife (permit number: 08487).

## DNA Deposition

The following information was supplied regarding the deposition of DNA sequences:

The raw transcriptome data for this project are available at NCBI SRA: PRJNA1152110; SAMN43336009–SAMN43336036.

## Data Availability

The scripts are available at GitHub and Zenodo:

– https://github.com/nicolemejia6180/Scripts-for-Wedge-Tailed-Shearwater-Plastic-Analyses.

– Mejia, N. (2024). Scripts for Wedge-Tailed Shearwater Plastic Analyses. Zenodo. https://doi.org/10.5281/zenodo.13621264.

## Supplemental Information

Supplemental information for this article can be found online at http://dx.doi.org/10.7717/peerj.18566#supplemental-information.

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
