# Peer review of "Effects of plastic ingestion on blood chemistry, gene expression and body condition in wedge-tailed shearwaters (*Ardenna pacifica*)"

_PeerJ, doi:10.7717/peerj.18566_

## Round 0.1 · original submission · Major Revisions

Dear authors,

Thank you for submitting your manuscript to PeerJ, and for entrusting us with the opportunity to handle your work. After meticulous review and evaluation by three reviewers, we have reached a Major Revisions neeeded decision concerning your submission.

Reviewer 1 provided insightful feedback on the experimental design and the validity of your findings. They underscored the importance of clearly articulated hypotheses, well-structured methods, and accurate presentation of results in relation to existing literature. Their suggestions for revisions aim to enhance the coherence and flow of ideas throughout your manuscript.

Similarly, Reviewer 2 offered valuable insights, particularly regarding the interpretation of findings and the necessity for additional methodological details, particularly in regard to P values. They emphasized the need for objectivity in reporting statistically significant results and recommended revisions accordingly.

Reviewer 3 provided valuable feedback on the experimental design and validity of the findings. They emphasized the importance of presenting clear hypotheses and structuring the methods and results in relation to existing literature on plastic pollution. Additionally, they highlighted that most of the results did not directly relate to plastic ingestion but rather compared physiological metrics with each other. Reviewer 3 suggested revising the paper's introduction to better justify the testing of these metrics and proposed incorporating hypotheses and predictions to enhance the overall structure and coherence of the manuscript. These recommendations will be addressed in the revised version of the paper.

Additionally, reviewers raised minor points pertaining to various sections of your manuscript, including the abstract, introduction, methods, results, and discussion. Addressing these points will undoubtedly elevate the overall quality of your work.

In light of the reviewers' feedback and our own assessment, we cordially invite you to submit a revised version of your manuscript. We firmly believe that with the necessary revisions, your study holds the potential to make a substantial contribution to the field.

Please ensure that you thoroughly address all the reviewers' comments and provide a comprehensive, point-by-point response in a cover letter accompanying your revised manuscript. Additionally, clearly indicate the changes made in the revised manuscript for ease of evaluation.

We eagerly anticipate receiving your revised manuscript.

Best regards,

Armando Sunny

Reviewer 1 ·

Basic reporting

This paper reported a study about the sublethal effects of plastic in seabird Ardenna pacifica on Hawaii. This paper is clear and unambiguous with professional English used throughout.

Experimental design

The original primary research is within aims and scope of the journal.

Validity of the findings

The major impact and novelty are meaningful but please provide some additional information
1. The statistics of the size of the found plastics. Since the introduction part mentioned the effects of micro-plastics but the title has plastic ingestion. It is better to clarify whether this is to study the effects of micro-plastics or plastics.
2. the p value in figure 2 cannot be concluded as significant since p < 0.05 is a minimal standard.

·

Basic reporting

This study reports on relationships between the presence of plastic (positive/negative from gastric lavage) in adult wedge-tailed shearwaters in Hawai’I and a wide spectrum of measurements relating to morphometrics, blood chemistry and gene expression. This is an interesting study that has made some questionable analysis/interpretation decisions. However, it addresses an interesting and novel topic of substantial public interest and I believe that it can be brought up to standard with major revisions.

I was very excited about this study when I read the abstract – given shearwaters high exposure to plastic, breeding in such close proximity to the great pacific garbage patch. What I took away from a thorough review of the results of this study is that there were no statistically significant relationships relating plastic presence to morphometric/blood chem parameters / gene expression using P<0.05. However, this is not the impression the abstract gives, which describes a litany of negative relationships between plastic and sub-lethal health impacts. In my view this way the results are presented is deliberately misleading, given that many researchers (unfortunately) don’t read past the abstracts and then unintentionally propagate misinformation. A substantial scientific quality is the unsubstantiated decision to redefine what counts as statistical significant p-value. Not finding statistically significant (P<0.05) relationships is a valid result and can be discussed in context (especially when there’s relationships of P<0.1 and low sample sizes – an argument can be made for further study to rule out type 2 errors). So, in essence, I agree that P<0.1 relationships are noteworthy and warrant further investigation, but it is a slippery slope if scientists start picking and choosing significant P values to suit a particular narrative.

Another important issue is the lack of detail in methods about how birds were selected. This could be a simple omission that needs adding or could completely confound the results depending on how it was done and the time-frames between sampling the birds used in this study. Please add this information about when birds were sampled, whether they were breeders/non-breeders and how the sampling regime relates to the chick-rearing period on these islands.

Aside from these major issues, the scientific writing was poor quality in sections for an international journal such as PeerJ. I am assuming that this is a student’s work based on the writing style (especially in the introduction and discussion), and if so, this manuscript would greatly benefit from the supervisory team / more experienced researchers having stronger editorial oversight before submitting this work for publication.

Just a personal note from the reviewer- Plastic pollution is a major global issue and undoubtedly a threat to many species. However, this scientific field also suffers from what I consider “alarmist” interpretations of results by some research teams, contrasted with objective reporting from others. I believe that these more alarmist interpretations are probably well-meaning and coming from a place of stressing urgency to address an environmental issue. However, this style of reporting can mislead scientists, decision makers and the public with respect to how well-evidenced the impacts of which threats (plastic or otherwise) are on which species. This can impact threat prioritization and redirect valuable resources away from the more significant threats to biodiversity. I implore the authors to please interpret their results with the appropriate level of objectivity that would be expected in other fields.

Experimental design

The experimental design was suitable for the questions asked, but I think this sample size may be too small (evidenced by some results that may in fact be statistically significant but type 2 errors cannot be ruled out). There’s also some details missing from method that make it hard to review the appropriateness of the sampling design of how birds were chosen. However, it is an interesting first investigation and worth publishing.

Validity of the findings

The findings have been interpreted in a misleading way. None of the plastic-health relationship described in the abstract were statistically significant, however those with P<0.1 warrant further investigation given small sample sizes. The body mass – health relationships described were statistically significant and warrant further discussion. There are large sections that need to be rewritten to objectively reflect the validity of the results, but it is worth publishing when this is corrected.

Additional comments

Minor comments:
Abstract
This abstract should be re-written to present these results with objective reporting of the statistical significance of these findings. It is fine to state the relationships detected that were near significant provided it is clearly stated.
53 – WTSH not WTHS

Introduction
References - There are missing references throughout (both unreferenced sentences and citations in-text that did not appear in the reference list), some references misrepresented / showed selective interpretations of the authors study, and there were several examples where the author cited is not (or probably not) the primary source. Please carefully check references before resubmitting as this should have been dealt with long before this got to review.
Note I didn’t check every single reference – but found enough examples of the below where I am familiar with the papers that I recommend the authors check each reference to make sure it faithfully represents the article.
67: REF
68: Didn’t see NOAA 2022 in the reference list so I couldn’t verify, but I suspect that NOAA may be referencing a primary literature rather than the primary source of this statistic. There are more up-to-date primary literature than projections based on 2020 than this that I recommend prioritizing.
72: REF
76: rather than use the generic word ‘amass’ and ‘toxins’, which can be interpreted several ways, I would recommend being specific here. Such as “adsorb persistent organic pollutants” or similar (haven’t read Verla 2019 thoroughly, but whatever compounds are actually discussed)
79: REF
80: REF
81-82: While not strictly incorrect in specific scenarios, this is a rather selective interpretation of this paper. I recommend re-reading Koelmans et al. 2016, and some of his more recent work, and report these conclusions here more faithfully with the actual results of this very well-respected study.
85: Where is Lavers 2019 in your reference list? Is this paper the primary source of this statistic (I suspect not).
86: REF
88: Gallo et al. 2018 is a review. It’s fine to cite a review but you should also cite a primary literature when the review author did not make this discovery.
90: REF? You can’t stay “studies suggest” and then not cite any…
95: Same comment as for 88
96: As above
98: mutagenic what?
103: This is a misrepresentation of Thiel et al. 2018’s study. Also, Thiel et al. 2018 is not in the reference list.
105: Wang et al. 2021 is a review. Like with the comment for Gallo, it is good practice to also cite the primary literature for these findings as well as the review. For example, imagine no one cites your primary study in the future, and instead only cites a future review article (Wedge-tailed shearwaters blood chemistry and plastic blah blah blah (Someone else’s review et al. 2026)) to describe your primary findings, without also citing this work? Funding for many scientists is influenced by citations so it’s good practice to support important primary research by attributing the original work to the correct author.
105 – 108: Lavers 2019 is not in the reference list. Also see the response / comment article to this study Roman, L., Gilardi, K., Lowenstine, L., Hardesty, B. D., & Wilcox, C. (2020). The need for attention to confirmation bias and confounding in the field of plastic pollution and wildlife impacts: comment on “clinical pathology of plastic ingestion in marine birds and relationships with blood chemistry”. Environmental Science & Technology, 55(1), 801-804.
112: REF
122 – 124: It is predicted by a 1993 study? How about all of the studies in the past 30 years that have actually studied/reported plastic ingestion in wedge-tailed shearwaters in many difference oceans?
126: NOAA what? Where is this in your reference list?
130 – 139 : this is more methods in terms of justifying which measures you used and why
140: this aim is far too general. Be specific to your work.

Methods
144-145: please list the sites and GPS location in the methods text.
Methods – when did you visit these study sites? Were all the samples collected in single trips? Multiple trips? What dates? How did this relate to breeding? Were these breeding or non-breeding birds? Were they caught as soon as the bird returned or after offloading the meal to the chick? Given that shearwaters offload plastic to chicks with meals, the plastic retrieved on lavage is going to be very strongly influenced by when this sampling was done both with respect to the chick rearing period (breeding/non-breeding and before, during or after) and if during, whether before or after feeding the chick.
150 – 162: this whole section is messy and includes a mix of methods/results/discussion. Please sort this out with your co-authors and place these in the correct sections.
164: how many times was each bird lavaged? Temp of the water? Anything given after (supp feed, electrolytes etc?) Please include details.
167 – 168: how many birds on which different islands? 29 total?
171: if blood sampling happened before lavage, please order the methods accordingly
183: Recommend improve logical order by separating the blood sampling into different subheadings to separate the different processes (for example the lab processed from the field processes). Within these lab processes section, can subheading for the different procedures – bird sexing, iSTAT (was this done in the field or the lab? We usually use iSTAT in the field with whole blood but here it sounds like iSTAT was done later with heparinised blood in the lab? Please clarify in methods). RNA isolation is in its own section, which is good, but I would move this along with the other lab procedures (all after the morphometrics, which also presumably happened in the field).
236-237: “Differences were considered statistically significant when p < 0.07.” Why? This is a decision that is outside of scientific norms and warrants a very good justification.

Results
254: Presence/absence is a first step, but most modern plastic ingestion papers require a higher level of detail about the quantity and nature of the plastics recorded. Please also add a description of the plastic (how many items, what material, mass etc following Provencher et al. 2019 (Provencher, J. F., Borrelle, S. B., Bond, A. L., Lavers, J. L., Van Franeker, J. A., Kühn, S., ... & Mallory, M. L. (2019). Recommended best practices for plastic and litter ingestion studies in marine birds: Collection, processing, and reporting. Facets, 4(1), 111-130.) Photographs of the items would also be good in the SI if possible (if you retained these items).
262: What do you mean by “minimal relationship”? Do you mean that you found no statistically significant relationships?
262 – 265: Describing results as “measurements above” is confusing for readers. Please present these results clearly and concisely. A nice way to present these results might be (and I would recommend to present it this way, though if you have an alternative way to clearly present it, please feel free to do so) in a table with the variables tested in the columns (for example a “dependent variable” column might have variables such as “Wing chord”, “TCO2” etc and a predictor variable column might say “Plastic presence” or “Bird sex”) and then a column stating the test done (For example “t-test”), a column for the sample size (n=27) and then a column for the statistical outputs, include the P-value. You can star (*) the significant test results.
269 – 270: this is not true, WTSH male morphometric measurements and mass is larger than females, on average, though there is overlap in the middle (smaller males / larger females, much like humans). Any large dataset of WTSH morphometrics shows this. You sample size of 10 male and 17 female birds might not be high enough to robustly pick up differences (Type 2 error, which may also be the case for some other results but will address later in discussion comments).
271: What do you mean by “Weight deviated from the other variables in the presence or absence of plastic under PCA 3 and 4?”
271 – 274: GLMs are typically used to compare two continuous variables, but here the results describe a continuous variable (weight) with a discrete variable (plastic presence/absence), also as shown in Fig 5a, which is an interesting choice. Is the continuous variable parametric or non-parametric? T-test or Mann-Whitney would be the more common test in this situation. Irrespective, going with the GLM, the p-values here shown in fig 5a are P=0.086 and P=0.073. While noteworthy, neither of these are statistically significant, not by the standard P<0.05 nor by the P<0.07 described in methods line 236-237.
272 – 274: Important to state that this relationship, while noteworthy given P<0.1, however, was not statistically significant and include the model output.
274-272 & Fig 6: Though not what you’re seeking to find within your aims, these relationships between body mass and BUN, hematocrit and potassium are actually statistically significant (P<0.05) and this is an important contextual outcome for your study.
297 – 303: please include n= next to these analysis when you’re examining by sex so the readers can make a judgement about significance based on sample size, which I am guessing is quite low for some of these. Please also be clear what the P-values were as it is unclear whether these relationships are statistically significant.
304-305: on what basis were these weight categories assigned? What are the cutoffs? Is this based on any other data showing WTSH mass during breeding for example (assuming these samples were collected across one trip. If they were collected at different times of year and in different trips the plastic status could well be related to chick offloading rather than exposure).
305 – 308 “Birds that did have ingested plastic, tended to be heavier” the way this is written is confusing, at first I read that birds with plastic are heavier. Please edit for clarity.

Discussion
General comments
As for the abstract recommendation, I recommend completely re-writing this discussion to discuss these results without misrepresenting the statistical significance of these findings. To me this discussion looks like a student assessment piece that has been submitted for publication with little change to the content. As it stands, this discussion section is very speculative, makes tenuous links hinging on chains of logical leaps to explain results that were not statistically significant in the first place. For a student assessment these sorts of ideas are encouraged because they demonstrate to the assessor that the student has read widely and thought about the problem. However, for a scientific publication, many of these links are far too speculative, and the discussion should instead focus on what the research actually found, how this fits into the context of other research (especially other seabird-plastic research) with well-supported similarities / differences between your work and others. And then future directions – what needs to be done next?
As it currently stands, this discussion overlooks the actual key/novel findings, and I recommend the new version focus on these findings, which were:
1) This study found no significant relationships between plastic presence and the examined parameters;
2) This study did find significant relationships between body condition and some blood chemistry parameters;
3) HOWEVER, due to small sample size and some P<0.01 relationships between plastic – health parameters, type 2 error cannot be ruled out and further investigation is warranted on A, B and C.
Following this, I recommend having a heavier focus on what is written in the seabird / other bird literature too. There is a growing body of literature in this space now. What do other authors find? Are their interpretations justified? Are there any criticisms? Etc. After this, what needs to be done in the future to gather more empirical support to either confirm or reject the relationships noted.

Minor comments
319 -321: This is the first mention of sublethal effects. Sub-lethal implies a negative health impact. This argument can be made for parameters such as body mass, where there is a well-established link with animal fitness. However, for gene regulation, do we know enough to say that different = sublethal? (I am not an expert in gene regulation, but one point that is beginning to become raised in the aquatic animals pollutants research space – not just plastic but pollutants generally - (which is much better researched than seabirds because you can do many sorts of experiments in lab conditions) is that markers that are sometimes described as “health impacts” of pollutant are actually just markers of an exposure and not necessarily a health impact.
322 – 323 : microplastic load wasn’t examined, only presence/absence. Also, here it says microplastic but elsewhere it says plastic - please keep terminology consistent.
323 – 326: but none of these relationships were statistically significant, which is worth mentioning.
333-335: This is fine to discuss but given that you didn’t test any of your plastic for these compounds, and that these relationships in your study are not significant anyway, is this worth including? And if it is, please phrasing this accordingly.
338 – 340: A lot of work has been done in this space since 1987 and worth providing more up to date literature.
343: REF
343: on the contrary, this was more than a marginal relationship, this was a strongly statistically significant relationship (p=0.002) between bird mass and BUN. Though there was very high spread of data (though R2 not reported) showing there’s more going on here to drive BUN values than body mass alone, which you’d expect.
345: REF
346: Do you really think so? This seems like a logical leap especially since the plastic results are not statistically significant. I would think that these results are showing that BUN and other blood chemistry parameters are more heavily being influenced by the birds’ body condition / mass (given its statistically significant) plus a whole bunch of other unknown factors. Knowing what time of the breeding cycle the samples were taken would also be useful context. For example, if they’re incubating, they go through long periods of food absence.
351 – 353: Is this worth including? This is also a chain of logical leaps when you didn’t actually measure BPA in either the plastic or the birds blood, nor did you test estrogen levels in any of the birds. Especially since the authors suggesting that males would present high estrogen levels, this is a big statement hinging on very little. Also noteworthy is the actual plastic loads aren’t presented. If a bird produced one tiny 1mm2 item of plastic, do you really think that would be enough to cause detectable changes in estrogen? Feeding experiments where quail were fed plastic in lab conditions, and estrogen measured, show no significant differences.
355-356 – I would suggest that the first step would be establishing whether this relationship is even significant, and THEN look at sex hormones. By the way, sex hormones and ingested plastic has been tested experimentally in birds (Japanese quail) across multiple studies, noting the loads that these birds are exposed are likely higher than that demonstrated here (based on the photo).
386 – 369 – yes, the false negatives due to type 2 error is well worth exploringand elaborating on. And equally to this is discussion of potential false positives based on sampling design (hard to judge without extra method detail) and data-dredging approaches, whereby by random chance alone you could expect one P>0.05 for every 20 relationships tested. Both these things equally merit discussion in this section.
369 – 375: this caveats section is good and worth elaborating on. Quite a few of these maybes have actually been tested in other studies, and you can find some literature to support and better flesh out these caveats.
378 – 379 – What do other studies find? There’s a lot of literature on this for seabirds, including studies that found negative, positive and no relationship between plastic and bird body mass. It’s worth contextualizing your study, and this P<0.1 relationship, among these other studies.
Conclusions -please re-write this with consideration of all comments provided in this review.

Reviewer 3 ·

Basic reporting

No comment

Experimental design

This paper would be much easier to follow and understand if the authors presented hypotheses they were testing, and structured their methods and results in relation to predictions based on the literature to date. There is a plethora of work on plastic pollution, why are the authors not engaging with this work to make informed predictions about the suite of metrics they are testing with plastic ingestion?

Validity of the findings

Most of the results here don’t relate to the plastic ingestion, but just compares physiological metrics with physiological metrics. There is no rationale presented in the introduction that would warrant testing these things in relation to each other. Therefore, I suggest that the authors revise the paper’s introduction to make the paper’s ideas flow from the methods to the results to the discussion. Currently, this reads as a giant fishing expedition with the data just tested in every which way in relation to each other. A set of hypotheses and predictions would help structure the ideas, why the tests were carried out, and what it all means in relation to each other.

Additional comments

Abstract – the line that states there are 43 genes that differentiate between males and females is unclear. Are you stating that the gene expression is different, or that there are differing genes, which is more of a herediary study… the results are not clear in the abstract.
Abstract – the last sentence – We hope that….. – doesn’t really fit. That is a given in most studies… isn’t the hope that they contribute to future work? Also, this suggests that the data is open access, is it?
Line 67 – past tense, it was established….
Line 102 – This statement is not supported in the literature. Birds may ingest more plastic pollution than others, but this paper does not state that the effects are worse. Also, this paper is now 6 years old, and a huge amount of literature has been done since then. This statement needs to be revised.
Line 106 – avoid direct citations of statement when possible – instead try to summarise.
Line 126 – why is NOAA here?
In general, this study presents a huge amount of data, but lacks any hypotheses or predictions. Why where these metrics used? What was the expected direction in change in relation to plastic pollution?
Line 154 – why is mapping the plastic around the island a part of this report? There is no indication in the introduction that mapping is a part of this study? What is the purpose, how does it relate to the physiological metrics that are reported on?
Line 157 – ok, so you didn’t predict any differences, why?
Line 159 – the authors are mixing gene expression, and genetics concepts here. Populations differing in genes, i.e. genetic differences, is very different from differences in gene expression. The birds could have the exact same proportions of alleles, and very different gene expression in relation to xenobiotics… this language needs to be cleared up and more specific.
The authors present about a hundred endpoints in relation to plastic ingestion – how are they controlling for type II errors – if you keep testing things, you are likely to find a significant relationship at some point by chance.
line 254 - what do you mean by unidentified hard bits? Plastics? Anthropogenic? Fish parts? You need clarify whether you are using these in the plastics ingestion data.
Line 263 – you have listed a lot of metrics above, do you mean them all? Be more specific.
Line 272 – weight of the bird, or weight of the plastics?
Generally, all the splitting of the birds, with plastics, without plastics, by the sexes, lead to incredibly small sample sizes that are being tested. The authors need to consider this and how it relates to the results. Are the tests that are being employed sensitive to small sample sizes?
Line 317 – why is COVID-19 mentioned here. This is confusing, and given that birds are not susceptible to the virus type that COVID-19 is, this doesn’t make sense in the context of birds.

---

## Round 0.2 · Minor Revisions

Dear Authors,

Thank you for the excellent work in addressing the reviewers' comments. However, some minor revisions are still required, particularly editorial corrections and a more substantial revision of the results section, to bring the manuscript to an acceptable standard for publication.

We greatly appreciate your choice of PeerJ for submitting such an engaging and valuable manuscript.

Best regards,
Armando Sunny

Reviewer 1 ·

Basic reporting

Clear and unambiguous professional English used throughout

Experimental design

Original primary research within aims and scope of the journal

Validity of the findings

Conclusions are well stated, linked to original research question

·

Basic reporting

Dear authors and editorial team.
This manuscript has been improved substantially since the previous version, with the feedback given by the three reviewers carefully considered and taken into account in this revision. The resulting manuscript is much better quality than the previous. I reiterate that this is an interesting study and a significant undertaking for an undergraduate student and commend Nicole accordingly. The updated version presents a clearer, more thorough, balanced presentation of the results and overall interesting manuscript.
I recommend moderate revisions (in the middle of minor and major), however this time the revisions are primarily editorial rather than substantiative, given that the substantiative revisions were addressed well in the previous version. However, there are some small sections that require re-writing (such as the model output results section and consequent interpretation). Overall, good job and it is clear a lot of thought and effort has gone into revising this manuscript.

Experimental design

Experimental design is appropriate, though sample sizes small. However, this is well-noted throughout and caveated appropriately.

Validity of the findings

Much improved, and worth publishing after some further moderate revisions.

Additional comments

Abstract
47 affects –> occurs in (or similar)
48 ‘a’ mounting threat
50 documenting -> exploring/investigating the sublethal effects (or similar). Using the word ‘documenting’ in this way implies the authors are searching for evidence to support a pre-determined conclusion (not good science) rather than drawing a conclusion based on what evidence is observed (good science).
51 in -> ingested by (or similar. ‘in’ could mean lots of things)
53 documented -> explored/investigated (as above)
55-57 -> “We observed a negative relationship between body weight and plastic, although it was not statistically significant.” Not worth highlighting non-significant trends in the abstract (these can certainly be discussed in results/discussion however)
65-66 “Portions of this text were previously published as part of a preprint” is this meant to be here? I would suggest removing the pre-print once this manuscript is published

Introduction
106 “and we know of no studies of this kind on natural populations of seabirds” -> suggest rephrasing to something like “though gene expression in response to plastic exposure is more difficult to study in wild populations” (or similar).
110-111 “and have often been used to quantify levels of pollution, including heavy metals and other contaminants” this is confusing – seabirds are used to quantify heavy metals? I am not sure if this is a ‘seabirds as sentinels of marine contaminants’ type or statement or a ‘researchers have taken an interest in seabirds’ exposure to contaminants and consequent health impacts’ type of statement. I suggest rewording.
112 albatrosses are Procellariiformes.
115-118 “pollutants on overall health” is this statement about plastic, about plastic-associated contaminants, or pollutants generally?
145 write in full (rather than acronym) if the first word in a sentence
152 rather than ‘we sampled’ (which is a methods type statement), this could be rephrased to more of an aims type statement, such as “we aimed to investigate WTSH on Maui…..”. Rephrase throughout this paragraph so these reads as aims statements rather than methods statements.
Methods
General comments – there’s a figure for the amount of plastic found on beaches around Maui (1C) but this is no longer in methods. I agree with the other reviewer that said this isn’t necessary to add, but since it’s a figure, either add to methods (not recommended) or remove this 1C figure (recommended).
167-168 remove “Portions of this text were previously published as part of a preprint” part (and throughout as it appears a few times)
169 please provide estimated date range for egg laying period for those not familiar with WTSH phenology on Maui. Can be in brackets. In some colonies in Australia, WTSHs can be on the ground for four or more months doing pre-breeding activities prior to egg laying/incubation.
177 7 -> seven (write numbers ten and less in full)
178 10 -> ten
209-211 this is more of a discussion point for caveats, it’s fine to just state what you did in methods in my opinion. Taking morphometrics is standard practice, though you’re correct in that how well these translate to indices of body condition (and which pairings are best) is subject to debate and probably species dependant. But IMO that’s for your discussion section
212 and throughout paragraph -> recommend call this “Gastric lavage / water offloading for ingested plastic” or similar rather than gut sample testing (as gut sample testing is not very clear / descriptive).
222-223 “We focused on testing relationships between presence of plastic and body condition rather than load as an exploratory analysis.” This sentence does doesn’t go here as its more about your later analysis, not about how you did the gastric lavage
271-273 P<0.05 is not a conservative P-value – it is a widely accepted minimum for statistical significance, acknowledging all the ‘line in the sand’ arguments surrounding it. As reviewer 3 highlighted, given the sheer number of relationships examined, even P<0.05 is going to flag 5% of things tested just by chance. Increasing this to P<0.01 means you’re going to get 10% flagged as significant by random chance, and results must be treated cautiously with this in mind. I recommend rephrasing (and rearranging future arguments accordingly) to say something like “We considered P<0.05 as statistically significant and took note/reported also relationships with values P<0.1, given the relatively small sample sizes examined.”
281 normality testing happens before the modelling/statistical testing, so recommend mentioning it before the other tests
288-291 very long sentence. A GLM using binomial data is typically called a logistic regression.
Results
295 write in full (not acronym) if first word in sentence
300 gut samples – I recommend choosing more descriptive subheading (as for methods)
303 the plastic figure is 1B not 1C. You could also table 1 to your brackets.
307 I note earlier you didn’t measure/weigh these samples (mentioned in methods) but table 1 quantifies the items by count, which is still better-quality information than presence/absence alone. Please summarise this plastic count data in the text here (even if the fishing line bundles are counted as number of bundles and separated from hard fragment counts).
310 start by describing your raw results before describing the relationships between them. When it comes to describing the relationships, I recommend a new subheading, something like “relationships between x, y and z”
312-316 This section requires re-writing as this is not a standard way to present model output results. It’s fine to direct readers to table 2, but these results also need to be described in text also (as a side note, Table 2 requires more model output data than just the P-value so reviewers can understand the model outputs fully. Also, please keep the number of significant figures of these P-values, and anything else you add, consistent throughout. Sometimes the journal will specify how many significant figures it wants, but if it doesn’t, I recommend sticking with two significant figures).
Typically, significant (P<0.05) results only are presented in text, with their relevant model outputs, including P-values. Then it can be described that others were not statistically significant. In your methods you set out a framework for also describing results that fell between P=0.05 and P=0.01, therefore it would be appropriate to also mention these here (along with the relevant model outputs).
317-321 this is discussion on PCA results, please present PCA results in the typical way
322 – This is drawing the long bow with PCA results. The standard (and more robust) way to test whether plastic presence is related to body mass would be a logistic regression (GLM with family = binomial in R) between plastic presence and birds body mass, considering sex also in the model given males tend to be heavier than females. Which is what is described in line 325-327 (though not clear if sex is considered – it should be). If the GLM is significant, great!! Report it here and the model outputs in the results accordingly. If the GLM is not significant, then using a PCA and getting into the weeds of PC3&4 and over-interpreting them for a result you want to see is not a scientifically defensible way to go about this IMO.
Being brutally honest – the loads of plastic in your photo and described in table 1 are not big loads for a healthy (presumably – if it’s hanging around on Maui looking to breed) adult shearwater. I would be very surprised if you found a significant relationship between these sorts of plastic loads and body mass. Not saying it’s not possible, just my experience having worked on thousands of shearwaters across the years, including wedge-tails (and their ingested plastic), suggests there is a point at which you expect to find a relationship, and that point is much larger plastic loads than what I can see in fig 1B and described in table 1 (unless those fragments are huge!!).
323 is “weight” body mass? If so I recommend saying body weight or body mass rather than just ‘weight’ for clarity, as weight can mean a lot of things (especially when discussing models).
327-329 This is an interesting result and should have its own paragraph (rather than being a ‘however’ on the back of non-significant plastic relationships). Please report the model output in the results text.
330 - 345 this is not an area of expertise for me and I can’t really comment
346-356 some of this is methods and the framework should be laid out in methods section. Please report the model output as well as the P values. Report the significant relationships first then the P<0.1 relationships.
355-356 – how? Is this just by thirds? If so maybe rephrase to “lower third, middle and upper third” or similar.
360-361 please include model output

Discussion.
368 lower case ‘w’ for wedge-tailed shearwaters
370-372 don’t need to repeat methods
374-376 this repeats results a bit. A way to express these two sentences that is less repetitive of results might be “We found statistically significant relationships between body condition and some blood analytes, including negative relationship with hematocrit, blood urea nitrogen (BUN), and potassium.” Or similar
437 space needed between ‘from the’
437-439 – I would suggest this is extremely unlikely in the time of year you sampled, so unlikely I wouldn’t even raise it in discussion. Regurgitating is disadvantageous to the bird – because otherwise they’re throwing up all their undigested food too after a days-weeks foraging. I have never observed adult WTSHs regurgitating spontaneously in the pre-breeding season (unless due to stress when captured/handled, and even that’s uncommon – I have seen it twice from hundreds handled, once in the hand and once in the bag, both had huge undigested meals which may have been compressed on capture). They do regurgitate to feed chicks (but your birds are pre-breeders). Fledglings also sometimes regurgitate hard parts before leaving the colony. I’d hazard though this is unlikely unless you actually saw piles of wet, freshly regurgitated (not from last seasons fledglings) piles of squid beak and plastic around the colony.
450-453 – rephrase considering points about statistical significance raised earlier
463 – misspelled Puskic
465 – Both of these examples were growing young chicks and not adults. In fact, for the quails the effect disappeared when the birds reached adulthood, worth mentioning here because two very different life stages (and different taxa) compared to your study
Figures
1B and 1C figure label around the wrong way compared to image. Recommend removing the part about how much plastic was found around Hawaii as mentioned in method general comments.
Fig 3 – label the three images as A, B and C (B and C label missing). For B, I recommend reversing the colour so the lower P values are red and the higher P values are white. For A and C, arrange the x axis in either increasing or decreasing order (currently they’re High – Low – Medium).
Tables
As mentioned in results section, Table 2 requires more model output data than just the P-value so reviewers can understand the model outputs fully. Please keep the number of significant figures of these P-values (and anything else you add) consistent throughout. Sometimes the journal will specify how many significant figures it wants, but if it doesn’t, I recommend sticking with two significant figures).

---

## Round 0.3 · Minor Revisions

Dear Authors,

Following the reviewers' assessment, minor revisions are needed before your manuscript can be accepted. These revisions mainly involve clarifying certain aspects of the language and addressing a few questions regarding the validity of the findings. Thank you for the effort you've dedicated to this compelling manuscript, and I look forward to receiving your corrections soon.

Best regards,
Armando Sunny

·

Basic reporting

Well done, this is much improved and now publishable quality. I commend the authors - it was a long journey but the current manuscript is much stronger for the revisions. Novel and worth publishing.

Experimental design

Good though small sample size as appropriately acknowledged and caveated.

Validity of the findings

Good though small sample size as appropriately acknowledged and caveated / discussed.

Additional comments

General comments:
Well done, the authors have improved this manuscript markedly. It’s a nice piece now and I recommend accept, with only very minor edits (see below – mostly typos or editorial, I don’t feel need to re-review).
One minor comment is that throughout I recommend writing “WTSH” in full in text. Your study is just on one species and the name isn’t super long, so I don’t think there’s any need to abbreviate in the main document text.

Specific comments
Title: lower case ‘t’ for wedge-tailed (written Wedge-Tailed). I’d recommend lower case for the entire title, except proper nouns (and Ardenna in Ardenna pacifica) including the species common name but that’s more a style preference.
Abstract
53: WTHS -> WTSH
Introduction
Throughout – please include the scientific name of each species discussed at first mention.
96: seabass-fed -> is there meant to be a hyphen here? If discussing seabass a species please include scientific name as several fish species are called ‘seabass’
97: please add scientific name for Japanese medaka
98: zebrafish -> please add scientific name
118: gulls -> if one species, please include scientific name. If multiple, Larus sp.
121: “at least one” -> “another” (each is just one study)
124: Flesh-footed Shearwaters -> lower case ‘f’ for consistency and please include scientific name
149 (and elsewhere): write WTSH in full ‘wedge-tailed shearwater’, it’s not a huge name and there’s no need to abbreviate in the text and it just makes it confusing for readers.
151 – 163: Most of this could be methods
Methods
Looks good!
Results
313-314: AIC is only meaningful when the AIC of the null model is presented, which it isn’t. Recommend either add AIC of null model for comparison or remove AIC.
Discussion
393: lower case grasshopper sparrow
445-446: scientific name for chicks (these were domestic chickens) and Japanese quail
447: only in the case of the quail. For the chicken chicks, the experiment was ended before the birds matured.

Reviewer 3 ·

Basic reporting

There needs to be some language clarifications. For example, line 195 indicates that '25 birds for chemical analysis', but I think what the authors are referring to are biomarkers and essential elements. The term chemical, at least in the wildlife health literature, typically refers to contaminants. By using it here, it is confusing to the reader as I then looked for where the chemicals in the blood were being analyzed.

Experimental design

No comment

Validity of the findings

There are some confounding variables that the paper isn't really accounting for. The birds with more plastics had heavier masses. There findings that then report biomarkers in relation to mass, but one of the questions that is not addressed is what is the normal for these birds? And importantly, how do you tease apart the effects from ingested plastics to the effects of whatever is leading those birds to having higher mass. Typically higher mass is determined to be 'healthier', but we don't know what those levels are for most biomarkers.

---

## Round 0.4 · accepted · Accept

Dear Authors

We are pleased to inform you that your manuscript, titled "Mapping species of greatest conservation need and solar energy potential in the arid Southwest for future sustainable development", has been accepted for publication in PeerJ.

Thank you for choosing PeerJ as the platform to share your research. We look forward to seeing the positive impact of your findings in the academic community.

Best regards,

Armando Sunny